# Correlation between Phenotype and Genotype in CTNNB1 Syndrome: A Systematic Review of the Literature

**DOI:** 10.3390/ijms232012564

**Published:** 2022-10-19

**Authors:** Špela Miroševič, Shivang Khandelwal, Petra Sušjan, Nina Žakelj, David Gosar, Vida Forstnerič, Duško Lainšček, Roman Jerala, Damjan Osredkar

**Affiliations:** 1Department of Family Medicine, Medical Faculty Ljubljana, University of Ljubljana, 1000 Ljubljana, Slovenia; 2Department of Bioscience and Bioengineering, Indian Institute of Technology Jodhpur, Jodhpur 342027, Rajasthan, India; 3Department of Synthetic Biology and Immunology, National Institute of Chemistry, 1000 Ljubljana, Slovenia; 4Department of Infectious Diseases, University Medical Centre Ljubljana, 1000 Ljubljana, Slovenia; 5Department of Pediatric Neurology, University Children’s Hospital, University Medical Center Ljubljana, 1000 Ljubljana, Slovenia; 6EN-FIST Centre of Excellence, 1000 Ljubljana, Slovenia; 7Center for Developmental Neuroscience, Faculty of Medicine, University of Ljubljana, 1000 Ljubljana, Slovenia

**Keywords:** beta-catenin, loss of function mutation, intellectual disability, hypotonia, microcephaly, eye movement disorders

## Abstract

The CTNNB1 Syndrome is a rare neurodevelopmental disorder associated with developmental delay, intellectual disability, and delayed or absent speech. The aim of the present study is to systematically review the available data on the prevalence of clinical manifestations and to evaluate the correlation between phenotype and genotype in published cases of patients with CTNNB1 Syndrome. Studies were identified by systematic searches of four major databases. Information was collected on patients’ genetic mutations, prenatal and neonatal problems, head circumference, muscle tone, EEG and MRI results, dysmorphic features, eye abnormalities, early development, language and comprehension, behavioral characteristics, and additional clinical problems. In addition, the mutations were classified into five groups according to the severity of symptoms. The study showed wide genotypic and phenotypic variability in patients with CTNNB1 Syndrome. The most common moderate-severe phenotype manifested in facial dysmorphisms, microcephaly, various motor disabilities, language and cognitive impairments, and behavioral abnormalities (e.g., autistic-like or aggressive behavior). Nonsense and missense mutations occurring in exons 14 and 15 were classified in the normal clinical outcome category/group because they had presented an otherwise normal phenotype, except for eye abnormalities. A milder phenotype was also observed with missense and nonsense mutations in exon 13. The autosomal dominant CTNNB1 Syndrome encompasses a wide spectrum of clinical features, ranging from normal to severe. While mutations cannot be more generally categorized by location, it is generally observed that the C-terminal protein region (exons 13, 14, 15) correlates with a milder phenotype.

## 1. Introduction

CTNNB1 Syndrome is a severe autosomal dominant neurodevelopmental disorder usually caused by de novo loss-of-function mutations in the *CTNNB1* (Cadherin-associated protein, beta 1) gene [1]. CTNNB1 Syndrome manifests itself in a variety of developmental disorders including Neurodevelopmental Disorder with Spastic Diplegia and Visual Defects (NEDSDV), and visual disorders including Familial Exudative Vitreoretinopathy (FEVR). NEDSDV is a neurodevelopmental disorder characterized by global developmental delay, impaired intellectual development with absent or very limited speech, craniofacial anomalies and microencephaly, axial hypotonia, and spasticity [1,2]. FEVR is an autosomal dominant disorder characterized by incomplete development of the retinal vasculature [3]. De novo loss-of-function mutations in the *CTNNB1* gene were first discovered in 2012 after diagnostic exome sequencing of individuals with severe intellectual disability [4], and since then the term CTNNB1 Syndrome has become the generic term for all disorders associated with *CTNNB1* haploinsufficiency. Currently, this disorder is diagnosed in approximately 300 patients worldwide, although this number is likely an underestimation due to misdiagnosis in cerebral palsy [5,6], leading to efforts to reevaluate the diagnoses of cerebral palsy patients to enable genomics-based changes in their clinical care.

The *CTNNB1* gene is located on chromosome 3 (locus 3p22.1, 41240942–41281939). It consists of 16 exons, with exons 2–15 (2346 bp) providing the coding sequence for β-catenin protein. β-catenin protein consists of 781 amino acids and belongs to the armadillo family of structural proteins involved in both embryonic development and adult homeostasis where it plays two essential roles: (1) as a transcriptional co-factor in the Wnt-signaling pathway, and (2) as an anchor in intracellular contacts and cell adhesion [7]. When the Wnt pathway is not stimulated, most of the newly expressed β-catenin is depleted from the cytoplasm by the destruction complex, while the remaining undergraded β-catenin engages with E-cadherin and α-catenin in membrane complexes that serve as cellular anchors. Within the destruction complex, which consists of Axin, Adenomatous Polyposis Coli (APC), and CK140 proteins, β-catenin undergoes a series of consecutive phosphorylations by the glycogen synthase kinase 3β (GSK3β) and CK1 kinases, which ultimately leads to β-catenin ubiquitination by β-TrCP and its proteasomal degradation. In the course of the canonical Wnt signaling pathway, Wnt ligands bind the membrane Frizzled family receptor that stimulates Dishevelled protein to sequester destruction complex proteins [8]. In this way, the degradation of β-catenin is inhibited, allowing the accumulation of free β-catenin which, transported to the nucleus, assists the T-cell factor/lymphoid enhancer factor (TCF-LEF) family of transcription factors in the transcription of various developmental genes, such as axin 1 and cyclin D. Structural and signaling roles of β-catenin are mutually exclusive, which is reflected in its protein structure. β-catenin consists of three regions with a distinct charge distribution: (1) an unstructured N-terminal region (130 amino acids), bearing amino-acid residues important for β-catenin degradation (S33, S37, Y41, S45); (2) a highly conserved central core region (550 amino acids) consisting of 12 armadillo repeats (each is a 42 amino-acid triple helix) [9,10] that form a positively charged groove [9], where β-catenin interacts with more than 20 protein partners including E-cadherin, TCF and degradation complex proteins [10,11,12]; and (3), the unstructured C-terminal region (100 amino acids), which is believed to enhance β-catenin stability by shielding the N-terminus from the destruction complex [9,10,13]. The molecular mechanism of binding exclusivity for the various β-catenin partners remains elusive—it is thought that the occlusion of ligand binding may be achieved by back-folding of termini.

Given the low prevalence of CTNNB1 Syndrome and its relatively recent discovery, little is known about the effect of *CTNNB1* mutation type and exonic localization on the severity of clinical phenotypes. It is also not clear whether *CTNNB1* mutations are null (in which case the mutated transcripts undergo nonsense-mediated RNA decay—NMD [14], or are translated into non-functional proteins) or, on the other hand, cause a partial loss of protein function due to the presence of an incompletely functioning protein. Another type of mutation can lead to the expression of proteins that interfere with the normal function of the protein from the wild type allele. These so-called antimorphic or dominant-negative mutations (mutated transcripts escape NMD and translate into truncated variants with potentially deleterious effects on the function of the healthy allele) are rare; however, given the variability of CTNNB1 Syndrome-associated mutations in terms of type and location, a production of auto-inhibitory truncated variants cannot be ruled out.

De novo mutations of the *CTNNB1* gene have been associated with neurodevelopmental disorders, with cases of intellectual disability and speech delay [4]. Addressing these open questions through phenotype–genotype correlation studies is essential in order to develop targeted interventions and focused clinical care, specific to the mutational context in the affected individuals [15]. The availability of data for such studies has been aided by genomic microarray technology, which has tremendously changed diagnostic approaches in children with neurodevelopmental disorders. Genetic testing can identify the genetic etiology in approximately 40% of cases of cerebral palsy (CP) cases, particularly those diagnosed with autism spectrum disorders (ASD) and intellectual disability (ID) with no apparent causative factor related to CP [5,16]. Access to a large number of patients who have been reliably and systematically assessed is fundamental for understanding CTNNB1 Syndrome. For the first time, this study provides a systematic review of previously reported cases in which we a) analyze the prevalence of clinical manifestations, and b) classify mutations according to their type (missense/nonsense/frameshift/splicing), exonic location, associated clinical features, and disease severity. Based on the analysis of the collected data, the genotype–phenotype correlations for CTNNB1 Syndrome are explored in detail. These may serve as a classification standard for new case studies and as a reference for researchers working to develop personalized therapeutic approaches.

## 2. Methods

This systematic review’s methodology and presentation follow the Preferred Reporting Items for Systematic Reviews and Meta-Analyses (PRISMA) guidelines. All the records were managed using the Endnote software program Endnote X4 (Thompson/ISI ResearchSoft Berkley).

### 2.1. Search Strategy and Inclusion Criteria

We included studies reporting mutations in the *CTNNB1* gene associated with NEDSDV and FEVR. Studies were included if they reported germline *CTNNB1* mutations, regardless of the amount of detailed phenotypic data on these patients. Language was not a restriction. Reviews, meta-analyses, abstracts, or conference papers, as well as studies on cell and animal models, were excluded from the data analysis.

Studies were identified by searching the following electronic databases from January 2012 to October 2021: PubMed, EMBASE, Web of Science, and CINAHL. The year 2012 was chosen because this was the first time that de novo *CTNNB1* mutations were reported [4]. The following key terms were used: *CTNNB1* AND (de novo OR loss-of-function OR germline mutations OR novel mutations; see Table 1 for an example of the search strategy in PubMed). To identify all studies that were not found in the literature search, we also screened the bibliographic references of the retrieved studies and reviews.

### 2.2. Data Extraction

Researchers SM and SK thoroughly reviewed the papers. The following key information was extracted: genetic mutation (exon number, variant, amino acid change, and mutation type), prenatal-neonatal issues (intrauterine growth retardation, IUGR), feeding difficulties, APGAR score), head circumference (presence of microencephaly), presence of muscle tone abnormalities, EEG and MRI results (presence of seizures, structural changes in the brain), dysmorphic features (broad nasal tip, small alae nasi, long and/or flat philtrum, thin upper lip vermillion), eye issues (e.g., strabismus, hyperopia, FEVR), early development (sitting, crawling, walking with support, walking independently), verbal speech and language comprehension (severity of speech delay and level of language comprehension), and behavioral characteristics (e.g., autistic-like or aggressive behavior), and additional clinical issues were extracted when available. During the extractions of genetic information from reviewed studies, we observed several mistakes in reporting exon numbers. Thus, each piece of genetic information was double-checked and estimated from the paper reporting on the genomic organization of the human β-catenin gene [17].

### 2.3. Quality Assessment

Each case report included in this review was evaluated against the adopted validated tools reported in case reports/case series [18,19,20]. This criterion was adapted to the present research topic (see Table 2 for quality assessment). We included six categories: ‘mutation analysis’, ‘demographic data’, ‘clinical assessment’, ‘cognitive assessment’ and ‘neuroimaging’, and ’neurophysiological investigation’. The highest possible score was 8. Case reports with a score of 7 or more were considered to be high quality, reports with a score of 5 or more were considered to be of moderate quality, and reports with a score of 4 or less were considered to be low quality. No data were excluded from this review, although case reports scoring as low quality were only included in the Appendix A and excluded from the genotype–phenotype analysis.

### 2.4. Genotype-Phenotype Analysis

In analyzing the patients’ mutation data, we classified mutations into five groups, according to the severity of symptoms: ‘normal’, ‘mild’, ‘moderate’, ‘moderate-severe’, and ‘severe’. Because this is the first attempt to perform genotype–phenotype analysis, there is no previous literature available for this rare syndrome, so we could not refer to the existing literature. Thus, the classification was based mainly on the patient’s eye contact (present/absent), speech (Viking Speech Scale, Pennington, 2010) [21], and cognition (ID; present/absent). Motor development was assessed only for normal and mild phenotypes because motor development in the moderate and severe groups was less related to phenotype/symptom severity. The classification into five severity groups based on phenotypic and symptomatic characteristics formed the basis for the assessment of genotypic–phenotypic correlation (Table 3).

## 3. Results

The search strategy identified 1221 articles and one study was additionally included based on the searches in previous reviews. After removal of duplicates and articles that did not meet the inclusion criteria, 28 articles were included in the review (see Figure 1 for flow diagram) [1,2,4,5,7,22,23,24,25,26,27,28,29,30,31,32,33,34,35,36,37,38,39,40,41,42,43,44]. Of the 28 articles, data from 84 patients with a *CTNNB1* mutation were pooled and were available for analysis. Data from all 84 patients are described in Appendix A and in the text (see Section 3.1); however, only 35 patients with sufficient data were included in the main genotype–phenotype analysis.

### 3.1. Prevalence of Clinical Features

Clinical features are presented based on primary (>50%) and secondary criteria (20–49%) (see Table 4 and Figure 2 and Figure 3), which were established to distinguish between more common and less common features. Facial dysmorphism was one of the most commonly reported clinical features (>86.8% of cases), including small alae nasi, long and/or flat philtrum, thin upper lip vermillion, and broad nasal tip. The presence of microencephaly (occipitofrontal circumference (OFC) less than 3 SD) was noted in 73.7% of cases, while the majority of the remaining cases had an OFC smaller than average. Reported cases exhibited eye abnormalities, including strabismus (52.6%), FEVR (22.8%), hyperopia (14%), astigmatism (8.8%), myopia (5.8%), esotropia (5.3%), retinal detachment (3.8%), and optic atrophy (1.9%). Muscle tone abnormalities were found in the majority of the reported cases, including axial hypotonia (91.5%), peripheral spasticity (84.7%), and dystonia, which was reported in 11 cases and not systematically assessed.

An electroencephalogram (EEG) was performed and reported in 30 patients, of whom 27 reported normal EEG (90%), whereas three patients had abnormal EEG (e.g., diffuse fast background activity, epileptiform activity with a tendency to spread). In addition, one patient’s report described focal epilepsy; however, it was not clear whether an EEG had been performed [5]. Magnetic resonance imaging (MRI) results were available for 24 cases of which 20 reports were normal (83.3%). Abnormal results were reported in four cases. These included arachnoid cysts, an enlarged Sylvian sulcus, hypoplasia of the corpus callosum, osteolytic lesions, enlarged lateral ventricles, abnormal gyration of the temporal lobe, absence of the right fornix and a hypoplastic brainstem, delayed myelination in the frontal lobes, mild dilatation of the ventricles, and mild thinning of the corpus callosum.

The gross motor milestone “sitting” was reported for 27 cases, of which 21 (77.7%) had reached this milestone at the mean age of 16 months. The remaining six had not reached it at that time, however, only three of them were older (age > 30 months) and the others still had time to reach this milestone (age ≤ 15 months). Of the 40 reported cases, 24 (60%) were able to walk independently, although most of them had difficulties (e.g., ataxic, unstable gait, use of an orthosis to stabilize the ankle). The average age for reaching this milestone was 3.8 years (range 12 months to 8 years). Of the 46 reported cases for speech development (Appendix A), 14 cases were nonverbal (30.4%), 19 cases used few words (41.3%), eight cases were able to speak short sentences (17.4%), and three cases were able to speak complete sentences and were only mildly delayed (6.5%). Two cases had no language delays in speech and had been achieving age-appropriate speech language milestones. In the majority of reported cases, receptive language was significantly better than expressive language. Of 25 reported cases, 13 (52%) reported “good” language comprehension, 11 (44%) reported “basic” language comprehension, and one reported “poor” language comprehension.

In most cases, the behavior was problematic. This information was available for 42 cases. Seven cases (16.6%) were described as ‘friendly and sociable’ and ‘with a generally cheerful demeanor’. In other cases, behavioral difficulties such as aggression (47.6%) were noted, ten cases showed stereotypic behavior (23.8%) and nine cases were diagnosed with autism spectrum disorder (21.4%). Three cases were diagnosed with ADHD (7.1%) and two with schizophrenia (4.8%). Data from eight cases indicated sleep problems, either in infancy (difficulty falling asleep) or in toddlerhood (night-time laughing fits).

Additional clinical features were considered to be rare (two or fewer cases; see Appendix A): Scoliosis, osteogenesis imperfecta, blue sclera, sacral dimple, left clubfoot, increased dermatoglyphic whorls, type 1 diabetes, dysplastic bicuspid pulmonary valve, delayed bone age, absent left testis, brachydactyly, Achilles tendon contracture, abnormal lung growth, pulmonary hypertension, mild thumb adduction, eczema, bicoronal craniosynostosis, single supernumerary maxillary incisor, bilateral orchidopexy, syringomyelia, hypermobile joints, and glue ear.

### 3.2. Genotype

The *CTNNB1* gene is located on chromosome 3 and spans 40.94 kb wherein the coding region is 2345 nt in length encoding a protein of 781 amino acids. β-catenin belongs to the armadillo family of structural proteins and is composed of three regions: an unstructured N-terminal region, bearing amino-acid residues important for β-catenin degradation; a highly conserved central core region consisting of 12 armadillo repeats, where several important interaction regions of β-catenin with many different protein partners reside; and the unstructured C-terminal region (Figure 4 and Figure 5).

Our dataset of 84 patients diagnosed with CTNNB1 Syndrome shows that *CTNNB1* genetic mutations are scattered throughout the gene with the majority of mutations located in the central armadillo repeat region (75.3%). The remaining mutations are roughly equally distributed between the N-terminal domain (10.6%) and the C-terminal domain (14.1%) (Figure 6).

Most of the mutations leading to CTNNB1 Syndrome were nonsense mutations (47.6%), followed by frameshift mutations (34.5%), missense mutations (8.3%), splice mutations (7.1%), and complete gene deletions (2.4%) (Figure 5). For all six missense mutations, American College of Medical Genetics and Genomics (ACMG) scores were extracted from both ClinVar and the Human Gene Mutation Database (HGMD). This information was available for three of six cases. Mutations c.1163T > C and c.2128C > T are classified as mutations of ‘Uncertain Significance’ and mutation c.1723G > A is classified as ‘Pathogenic/Likely Pathogenic’. The occurrence of mutations was found in all exons, except exon 1 and 2. The most frequently reported mutations include mutations in exon 10 (c.1603C > T, p.R535*), exon 9 (c.1420C > T, p.R474*), exon 7 (c.998dupA, c.999C > G, c.999del causing p.Y333*), intron 5 (c.734 + 1G > T, c.734 + 1G > A, causing splice mutation), and exon 13 (c.2038_2041dup, p.S681*) (Figure 6). Specific mutations in exon 9 (c.1272_1275del, p.Ser425Thrfs*11, and p.Glu479Argfs*18) and exon 4 (c.283C > T, p.R95*) occurred twice. Notably, there were two patients with reported gross deletion of the entire gene (Figure 7 and Figure 8). Other mutations are listed in the Appendix A. All but six cases found in three articles were reported as de novo mutations [24,29,35].

### 3.3. Phenotype–Genotype Correlation Analysis

There were 35 case reports that sufficiently met the study criteria and were further selected for detailed genotype–phenotype correlation analysis. As described previously, patients were divided into five groups according to the severity of the phenotype (Table 5 and Table 6).

#### 3.3.1. Severe Phenotype

Five cases included in this category had presented a severe speech impairment (absent speech or being able to speak only two words) and poor or no eye contact. Mutations were observed in intron 5 (*n* = 3, splice site mutation) and exon 6 (*n* = 2, frameshift and nonsense). Compared with the other groups, this group had the lowest IQ, indicating severe ID (IQ < 70). These cases showed more ritualistic and autistic-like behavior along with auto-aggressive behavior. The ability to walk was reported in four cases, and three of them were able to walk independently. The majority of the ‘severe group’ had unremarkable brain MRI and EEG reports and presented with axial hypotonia, peripheral spasticity, facial dysmorphism, and eye abnormalities (FEVR = 2, strabismus = 2 and hyperopia = 1); see Appendix A). Of the three patients reported, two had microencephaly, and one female patient (c.755delTinsAAC, p.Leu252*) had an OFC in the 13th percentile.

#### 3.3.2. Moderate-Severe Phenotype

Thirteen patients with a moderate-severe phenotype had mutations in exon 4 (p.Tyr142Valfs*4), exon 8 (p.Leu388Pro), exon 9 (p.Ser425Thrfs*11, p.Arg449GlnfsTer24, p.Arg474Ter), exon 10 (p.Arg535*), exon 11 (p.Gln601*), and exon 12 (Glu642Argfs*6, p.Glu642Valfs*5). The main characteristics of this group were severe speech impairments (no speech/few words; 92.3%) and inability to walk independently (84.6%). One patient lost the ability to walk due to progressive spasticity. The majority of included cases had good language comprehension (six out of seven reported cases; 87.5%). Eleven reports included information on behavioral problems, including aggression (72.7%), tantrums (45.5%), and stereotypic behavior (36.4%). All reported cases showed facial dysmorphism, axial hypotonia, and peripheral spasticity. All but one case had normal brain MRI and EEG reports. One case with missense mutation reported epileptiform activity with a tendency to spread. Seven cases had microencephaly (53.8%), and the remaining six cases had head size smaller than the 20th percentile. Eight cases reported strabismus, and hyperopia, optic atrophy, and FEVR were each reported one time (Appendix A).

#### 3.3.3. Moderate Phenotype

Twelve patients were included in this category. Mutations were present throughout the gene in exon 3 (p.Gly34Asnfs*15, p.Gln78*), exon 4 (p.Arg95*, p.Tyr142Valfs*4), exon 5 (p.Gly236Argfs*35), exon 7 (p.Gln309*, p.Tyr333Ter), Intron 7, and exon 10 (p.Gln538*), and two patients had entire gene deletions. This group is characterized by higher walking ability: Ten cases were able to walk (83.3%), and two cases may still have had time to start walking (ages 3 and 4). Seven reported cases showed good speech development (ability to combine words in seven cases) and the remaining five cases had only recently started speaking and may improve over time. Comprehension was good in all reported cases. Five patients in this group were described as having a friendly and sociable personality (41.6%), five currently or in the past exhibited aggressive behavior (41.6%), two exhibited temper tantrums (16.6%), and two exhibited stereotypical behavior (16.6%). Axial hypotonia was present in all twelve cases, while peripheral spasticity was present in eight cases (75%). All cases presented with CTNNB1-related facial dysmorphism (broad nasal tip, small alae nasi, long and flat philtrum, thin upper lip vermillion), and all but one case with eye abnormalities (strabismus = 5, hyperopia = 3, hypermetropia = 1, and esotropia = 1). Five cases were found to have microencephaly, while the rest (*n* = 7) had OFC smaller than the 33th percentile. EEG was reported normal in all except one case (diffused fast background activity during episodes; p.Gln78*). MRI reports showed abnormal results in four cases (see Section 3.1 for details of abnormal results).

#### 3.3.4. Mild Phenotype

Three patients were included in this group and all had a nonsense mutation in exon 13 (p.Ser681*, p.Arg661*). Two reported cases were siblings who both had a similar phenotype: mild ID with good language comprehension, and mild expressive speech impairment, such that they were able to speak in complete sentences. Both siblings were able to sit before their first birthday and walk independently before their second birthday. The report for the third patient was similar (sitting at 11 months, walking independently at 30 months, and speaking their first words before age four). All of these cases had behavioral problems. The siblings displayed symptoms of ADHD, autism, anxiety, aggression, and frustration, and would occasionally self-harm. The non-sibling case exhibited ‘obsessional behavior’ (no further data available). Both siblings, clearly with borderline intellectual abilities or mild intellectual disability, could speak in sentences, and reportedly enjoyed social and verbal interaction with others. Their expressive speech was limited, while their language comprehension was better. Available reports (for siblings only) showed normal EEG and MRI scans. All three cases presented with axial hypotonia, and only the siblings showed peripheral spasticity. One of the siblings had an average head circumference, while the other two cases showed OFC of −3SD and −6SD, indicating microencephaly. All three patients showed facial dysmorphism, including thin upper lip vermilion; however, only data from the siblings include a more detailed report on craniofacial features: broad nasal tip small alae nasi, and long and flat philtrum. Both siblings presented with strabismus and myopia, while the third patient had no data on the presence of eye abnormalities.

#### 3.3.5. Normal Phenotype

Two reported cases, a missense mutation in exon 14 (c.2128C > T, p.Arg710Cys) and a nonsense mutation in exon 15 (c.2142_2157dup16, p.His720*) presented with the Retinal Vascular Condition FEVR, but were otherwise developing within the normal range.

## 4. Discussion

The present paper provides a comprehensive and up-to-date review of published cases of CTNNB1 Syndrome, an analysis of the prevalence of the most common symptoms, and a classification of *CTNNB1*-associated mutations according to the severity of their respective phenotypes (Table 5, Table 6 and Appendix A). Based on the availability of sufficient data, 35 patients were included in the analysis. While we acknowledge that the size of the analyzed cohort was too small to perform a statistically significant genotype–phenotype correlation, we believe that the classification performed in our study may nevertheless be informative for future correlation studies and provide the classification basis for further data collection and analysis.

The main finding of this paper is that there is substantial variability within genotypes and phenotypes of patients with CTNNB1 Syndrome. Regarding genotype, we found that mutations associated with CTNNB1 Syndrome are scattered throughout the coding sequence of the gene, with the exception of the first coding exon (exon 2), although we cannot exclude the possibility that pathological mutations also occur in this exon. In terms of phenotype, we were able to classify patients into a spectrum of disease severity (severe, severe-moderate, moderate, mild, and normal). Our analysis of the mutations available in each phenotype category suggested certain relationship between phenotype severity and mutation location and type. The majority of the mutations analyzed were associated with moderate or severe disease phenotypes, manifested by facial dysmorphisms, microcephaly, various motor disabilities, speech and cognitive impairments, and behavioral difficulties. From a biochemical perspective, this was expected because the critical interaction surface of β-catenin is large, extending from armadillo repeat 3 to 9 encoded in exons 5 to 10, respectively (Figure 3, Figure 4, Figure 5 and Figure 6).

Consistent with these expectations, the analyzed nonsense and missense mutations occurring in exons 14 and 15, which are part of the C-terminal domain of β-catenin, were classified in the normal clinical outcome as they presented with eye abnormalities only, and otherwise had a normal phenotype. Both mutations were found in all family members, which suggests that they were inherited rather than occurred de novo [35]. Interestingly, a classic ophthalmological feature of CTNNB1 Syndrome—FEVR—was found in many of these patients (Appendix A), suggesting that it may be caused by alterations at the uncharacterized C-terminal domain. Biochemically, nonsense mutations impose a premature stop codon that results in a truncated protein, while missense mutations lead to the substitution of a single amino-acid in the protein sequence, in which case the severity of the consequences depend on the structural integrity of the mutant protein and possible disruption of binding sites for interacting proteins. Lack of disease severity for mutations in exons 14–15 is most likely attributed to the fact that a large part of the protein, should the transcript circumvent nonsense-mediated decay, is transcribed and is likely to be, at least partially, functional. Such hypomorphic mutations have also been observed in other genes (e.g., DMD, APC) [45,46]. Furthermore, helix C, a critical structural motif of the C-terminal domain with a role in the co-transcriptional activity of β-catenin [47], is encoded by exon 13; therefore, it lies upstream of mutations in exons 14–15. Accordingly, patients with mutations in exon 13 exhibit an array of additional cognitive and motor impairments (Table 5). One of the reported nonsense mutations in exon 13 is a mutation of a tyrosine residue at position 654 (Y654; Appendix A), which, in its phosphorylated state, is directly linked to stabilizing helix C, and thus allows accessibility to co-activators of β-catenin-mediated transcription leading to nuclear localization of β-catenin [48]. However, the mutations in exon 13 (c.2038_2041dup, p.Ser681* and c.1981C > T, p.Arg661*) were at worst classified into the mild-moderate phenotype category, which can be attributed to the fact that, while important for signaling, helix C appears to be completely dispensable for the structural role of β-catenin in cell-cell adhesion [9].

An additional explanation for the milder phenotype in mutations in exons 14–15 could be that helix C, which is the most important feature of the C-terminal domain, critical for the co-transcriptional role of β-catenin [47], is encoded upstream, in exon 13. Accordingly, patients with mutations in this region exhibit a range of additional cognitive and motor impairments (Table 5). One of the reported nonsense mutations in exon 13 occurs at the Y654 residue (Appendix A), which, in its phosphorylated state, is directly linked to keeping helix C accessible to co-activators of β-catenin-mediated transcription, and is consequently responsible for the nuclear localization of β-catenin [48]. Still, the mutations in exon 13 (c.2038_2041dup, p.Ser681* and c.1981C > T, p.Arg661*) were at worst classified into the mild-moderate phenotype category, which can be attributed to the fact that, while important for signaling, helix C appears to be completely dispensable for the structural role of β-catenin in cell-cell adhesion [9].

In contrast to nonsense and missense mutations, we found that frameshift mutations in the C-terminal region can cause a severe disease phenotype. A patient with a frameshift mutation (c.2273delA, p.His758Leufs*30) in exon 15 (Appendix A) presented with severe intellectual disability and symptoms of autism spectrum disorder [38]. Frameshift mutations can lead to a variety of changes, from the introduction of a premature stop codon to a protein extension beyond its normal stop codon. The consequences of these changes are difficult to predict without experimental analysis; however, in general, such mutations are considered deleterious [49] because they can undermine the structural integrity of the entire protein. Furthermore, splicing mutations have also been found to cause severe or moderate disease. In our analysis, splice mutations were located at the 3′ acceptor splice sites of intron 5 and 7, with patients presenting mainly with severe cognitive disability and lack of eye contact, while the most robust characteristics of the cases categorized as moderate-severe included their inability to walk (exons from 8–12). Splicing mutations likely behave like frameshift mutations in that they can cause retention of the intron during the process of splicing, which can be detrimental for the reading frame. Interestingly, a pair of patients with identical mutations (c.2092_2096dup, p.Ile700Leufs*37) exhibited varied symptoms [24]. Thus, the influence of a so-called modifier gene and/or additional environmental factors may play an important role in disease manifestation.

An additional interesting observation was that gross gene deletions resulted in a moderately pathological phenotype. Considering that these types of mutations likely result in null activity of β-catenin, we would expect them to cause the most severe phenotype possible. However, because many point mutations from our review appear to lead to a more severe phenotype, this is a possible indicator of dominant-negative effects. Dominant-negative effects are a consequence of mutations that lead to the expression of truncated proteins that obstruct the normal function of the protein from the wild type allele. It has been reported that β-catenin truncations without a C-terminal domain, which lack transcriptional activity, nevertheless bind co-transcription activators, and thus interfere with their binding to the wild type variants [50].

The present study showed that mutations associated with CTNNB1 Syndrome are also found in exon 3, although mutations in this region are usually associated with cancer, because exon 3 bears important amino-acid residues for β-catenin degradation, such as the CK1-α phosphorylation site (S45), GSK3-β phosphorylation sites (S33, S37, and T41), and Fbw1 (D32 and G34). These are essential components of the inhibitory destruction complex that controls the levels of the free cytosolic β-catenin [51,52]. Missense mutations in these residues have been associated with colorectal cancers and other types of malignancies (e.g., melanoma, brain tumors) as they lead to the accumulation of nuclear β-catenin and tumorigenesis [53]. In our data, four CTNNB1 Syndrome-causing nonsense and frameshift mutations in exon 3 were reported (G34, E54, Q72, and Q78), which were associated with a moderate phenotype. There is no evidence that these mutations cause tumor growth and progression.

Our review could not confirm the results reported by Rossetti et al. (2021) that missense mutations are associated with vitreoretinopathy [22]. This condition was found in 13 cases (see Appendix A), with no correlation with mutation type; however, we observed a higher prevalence of this condition in the Chinese population (found in 69.2% of Chinese patients) [24,29,33,35,39,40]. The FEVR condition affects the retina by preventing blood vessels from forming at the edges of the retina. It is predicted that reduced β-catenin levels increase the probability of causing FEVR, although the exact pathophysiology is yet unclear [35]. This condition is progressive in nature. Thus, regular screening of the retina can prevent unnecessary vision loss in patients with CTNNB1 Syndrome.

Some methodological considerations should be taken into account. The proportion of incomplete clinical data is perhaps the most significant and troubling issue in the current study. Most of the included studies were case reports or small case series, so referral, selection, and publication biases could occur. The results of our systematic review of the genotype–phenotype correlation are based on the data reported in the previous studies. It is, therefore, necessary to validate these results in prospectively gathered data and cell models. Importantly, this systematic review makes a clear demarcation of the exon borders of β-catenin, based on the article of Nollet et al. 1996 [17], and can guide future studies. The organization of the clinical criteria based on the prevalence of the phenotypical findings may help neurologists to determine whether they will screen their patients for CTNNB1 Syndrome. Based on the phenotype, this gene can be added to the appropriate panels.

## 5. Conclusions

In conclusion, it was possible to observe at least some evidence of genotype-phenotype correlation between the type and position of pathogenic variants and clinical expression. Most importantly, we found that nonsense and missense mutations in exon 14 and 15 result in a functional protein with a neurotypical phenotype and ocular abnormalities only, whereas the nonsense and missense mutations on exon 13 result in a milder phenotype. Splice mutations in intron 6 and mutations at the exon can lead to a severe phenotype. Mutations between E8 and E12 could be associated with severe motor disabilities. Global developmental delays, speech impairments, craniofacial features, and eye problems are the commonly observed phenotypes in almost all reported mutations. Future implications for neurologists, researchers, and patient advocates include using the developed correlation to predict the patient’s potential phenotype.

## Figures and Tables

**Figure 1 ijms-23-12564-f001:**
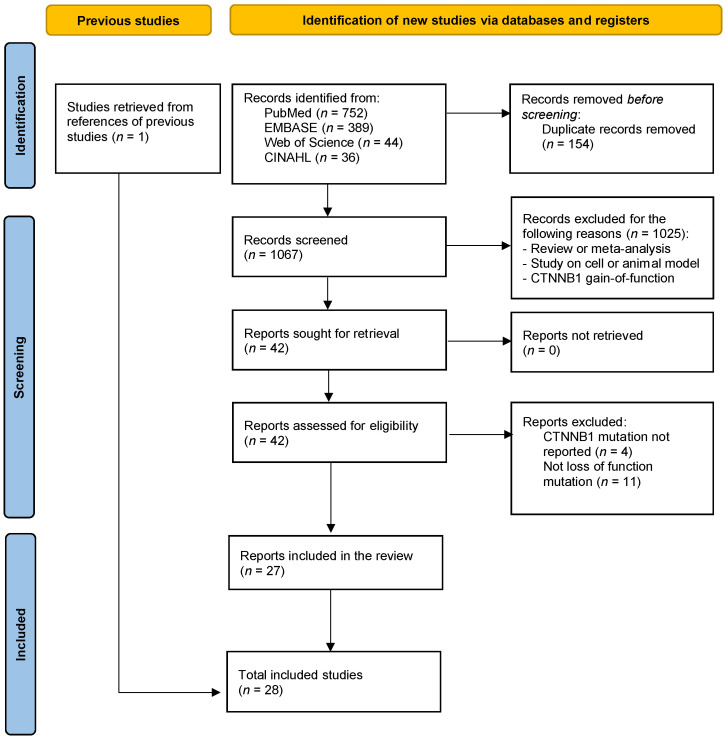
Flow diagram of the systematic review.

**Figure 2 ijms-23-12564-f002:**
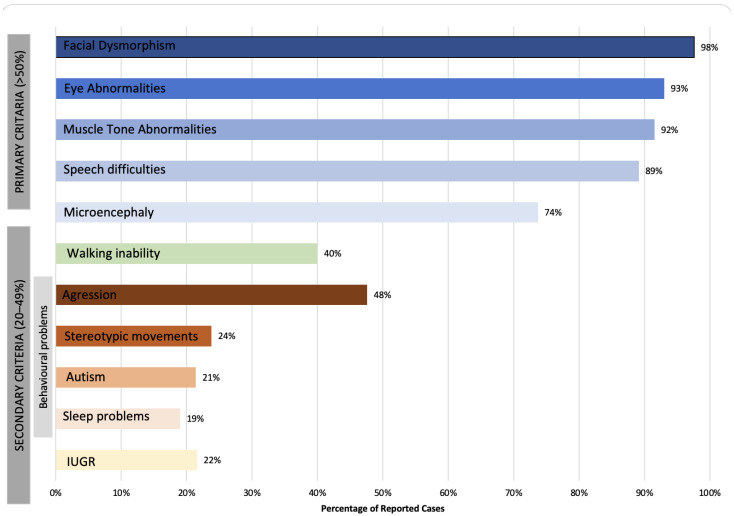
A 2-D bar chart representing clinical manifestations categorized based on the primary and secondary criteria.

**Figure 3 ijms-23-12564-f003:**
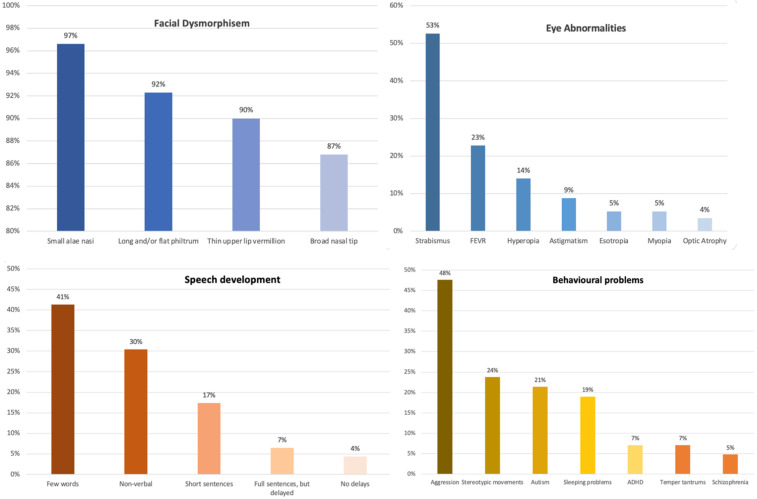
A column chart representing percentages of facial dysmorphism, eye abnormalities, speech development, and behavioral problems found in reported CTNNB1 patients.

**Figure 4 ijms-23-12564-f004:**
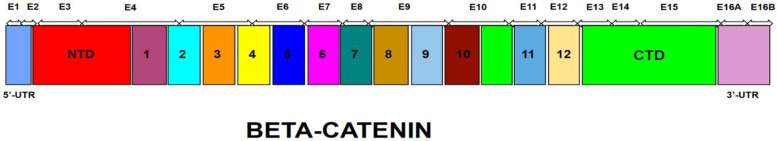
Schematic representation of the β catenin coding region; exon structure in correspondence to encoded protein domains.

**Figure 5 ijms-23-12564-f005:**
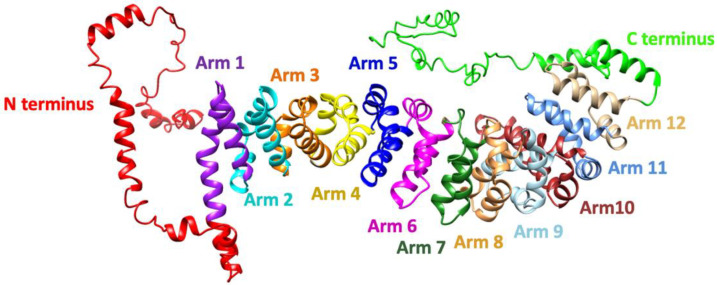
Three-dimensional model of human β-catenin protein generated by I-Tasser. Annotation was performed according to Huber et al., 1997. The model shows the N-terminus (red), armadillo repeat arms 1–12, and a helix and unstructured region of the C-terminal domain.

**Figure 6 ijms-23-12564-f006:**
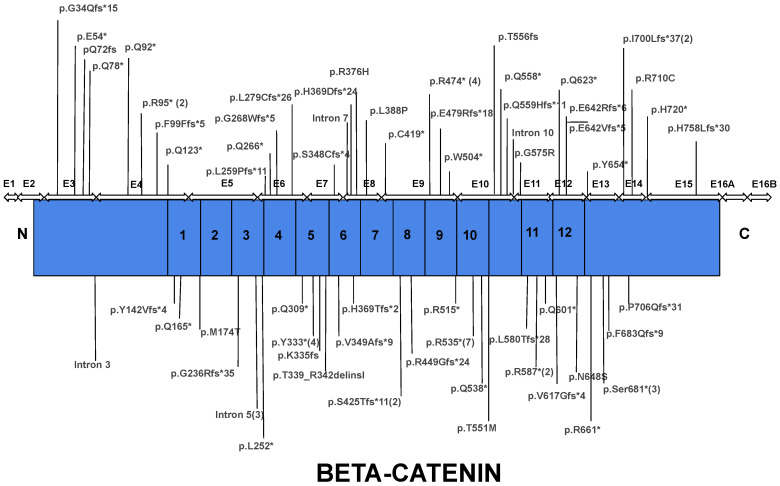
Distribution of mutation causative of CTNNB1 Syndrome throughout the protein coding region in accordance with exon location and subsequent encoded protein domains. Most mutations reside within the armadillo repeat region of the protein. Asterisk indicates a nonsense mutation and number in parenthesis indicates the number of cases reported the mutation.

**Figure 7 ijms-23-12564-f007:**
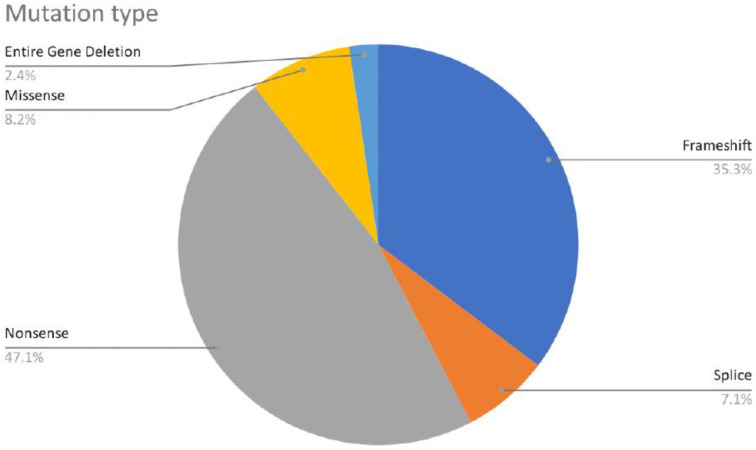
Distribution of CTNNB1 mutation types.

**Figure 8 ijms-23-12564-f008:**
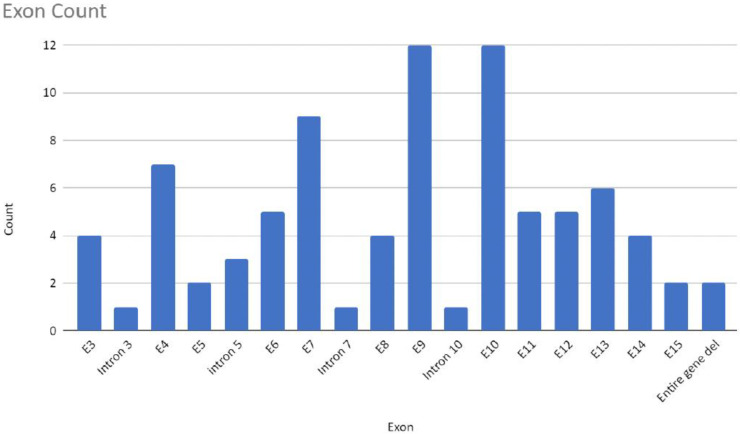
Number of mutations detected in individual intron or exon regions or whole gene deletions of analyzed samples in this study (n = 84).

**Table 1 ijms-23-12564-t001:** Search strategy in PubMed.

Number	Search Strategy
#1	*CTNNB1* [Text Word]
#2	*CTNNB1* Protein, Xenopus [MeSH Terms]
#3	#1 OR #2
#2	de novo OR loss-of-function OR germline mutation OR novel mutation [Text Word]
#3	de novo OR novel mutation [Text Word]
#4	loss of function mutation [MeSH Terms]
#5	germ-line mutation [MeSH Terms]
#6	#3 OR #4 OR #5
#7	#2 OR #6
#8	#3 OR #7

**Table 2 ijms-23-12564-t002:** Dataset for assessing quality of genotype–phenotype correlational studies.

Category		Points
Mutational analysis	None	0
	Prescreening methods	1
	Full sequencing	2
Demographic data	None	0
	Sex, age, (ethnicity)	1
Clinical assessment	No	0
	Neurological assessment (facial dysmorphism, achieving motor milestones)	1
	Neurological and cognitive assessment	2
Cognitive assessment	None	0
	Diagnostic test done (autism, IQ tests)	1
Brain screening tests	None	0
	MR	1
	MR and EEG	2

**Table 3 ijms-23-12564-t003:** Categorization of the severity for the genotype/phenotype analysis.

Variable/Severity	Eye Contact	Speech	Cognition (ID)	Motor Development
Normal	+	No delays	Normal ID	No delays
Mild	+	Delayed; speaking in full sentences	mild ID (55–70)	sitting before 1 year, walking before 2 years
Mild-Moderate	+	Delayed; can speak in sentences, speaking can be unclear	mild-moderate ID (40–70)	Sitting and walking independently with difficulties (ataxic)
Moderate	+	Delayed, speaking in simple sentences, can be unclear	mild-moderate ID (40–70)	Could be sitting and walking but with difficulties
Moderate-Severe	+	Simple words/no words; uses sign language	moderate-severe ID (20–55)	Could be sitting and walking but with difficulties
Severe	-	No speech	severe ID (20–40)	Could be sitting and walking but with difficulties

Notes. + present; - absent.

**Table 4 ijms-23-12564-t004:** Summarization of the clinical features categorized according to their prevalence (*n*= 84).

Clinical Feature	*n* (%)	Clinical Features	*n* (%)
*Primary Criteria (>50%)*		*Secondary Criteria (20–49%)*	
Presence of microencephaly (valid cases: 57)	42 (73.7)	Walking inability (valid cases: 40)	16 (40)
Eye abnormalities (valid cases: 57) ^1^	53 (93)	Aggression	20 (47.6)
Strabismus	30 (52.6)	Stereotypic movements	10 (23.8)
FEVR	13 (22.8)	Autism	9 (21.4)
Hyperopia	8 (14)	Sleep problems	8 (19)
Astigmatism	5 (8.8)	ADHD	3 (7.1)
Esotropia	3 (5.3)	Temper tantrums	3 (7.1)
Myopia	3 (5.3)	Schizophrenia	2 (4.8)
Speech difficulties (valid cases: 46)	41 (89.1)	Abnormal MR (valid cases: 24) ^2^	4 (16.7)
Non-verbal	14 (30.4)	IGR (valid cases: 37)	8 (21.6)
A few words	19 (41.3)	Additional criteria	
Short sentences	8 (17.4)	Scoliosis (not systematically assessed)	2
Full sentences, but delayed	3 (6.5)	Feeding problems (not systematically assessed)	5
No delays	2 (4.4)		

^1^ Patient can have several eye abnormalities at the same time; ^2^ dilated ventricles, underdevelopment of the corpus callosum and brainstem, delayed myelination.

**Table 5 ijms-23-12564-t005:** Categorized CTNNB1 cases according to the severity of the phenotype (*n* = 35).

Genetic Mutation			Gender	Age (yrs)	Facial Dysmorphism	Eye Conditions	Microencephaly	Axial Hypotonia/Spasticity	Achieving Milestones				Behavior and IQ
Exon no. and Variant	Amino Acid Change	Mutation Type							Sitting (mo)	Crawling (mo)	Walking Independently	Speaking	
SEVERE PHENOTYPE													
I5, c.734 + 1G > T	Splice mutation	Splice	F	32	+	Strabismus	+	+/+	2–5 years (40mo)	NA	No	Absent speech	Ritualistic behaviours with temper tantrums, autism, severe ID (18 months)
I5, c.734 + 1G > A	Splice mutation	Splice	F	49	+	FEVR	NA	+/+	NA	NA	NA	Absent speech	NA, IQ = 40
I5, c.734 + 1G > A	Splice mutation	Splice	F	27	+	FEVR	NA	NA	NA	NA	Walking at 49 yrs (ataxic)	Absent speech	NA, IQ = 20
E6, c.755delTinsAAC	p.Leu252*	Nonsense	F	15.3	+	Strabismus, hyperopia	-	+/+	NA	NA	10 yrs	2 words	Auto-aggressive behavior, stereotypic movements, short eye contact; severe IQ
E6, c.799_809delGAAGGAGCTAAinsGAA	p. Gly268TrpfsTer5	Frameshift	F	7	+	NA	+	+/+	18	NA	3 (broad based gait)	No speech	Autism, ID
MODERATE-SEVERE PHENOTYPE													
E4, c.423_424insG	p. Tyr142Valfs*4	Frameshift	F	5.6	+	Strabismus	+	+/-	NA	24	not yet	Severe, few words (30), sign language	Repetitive movements, ID
E8, c.1163T > C	p.Leu388Pro	Missense	F	6.8	+	NA	+	+/+	13	18	2.5	First word at 2 ½, 20 words at 4 years but not intelligible	ID
E9, c.1272_1275del	p.Ser425Thrfs*11	Frameshift	F	29	+	NA	+	+/+	24	3 years	8 (progressive spasticity now with support)	Started speaking first words 9–10 years; now able to speak a few words	Aggression, auto-mutilation, and fecal smearing
E9, c.1272_1275del	p.Ser425Thrfs*11	Frameshift	F	3.25	+	Strabismus	-	+/+	NA	NA	not yet	Babbles now, some words are understandable	Very happy and friendly, low frustration tolerance
E9, c. 1344_1345 InsertionA	p.Arg449GlnfsTer24	Frameshift	M	8	NA	Strabismus	+	NA/+	NA	NA	8	First words at 3 years, at 8 years able to speak short sentences	Aggression sometimes when frustrated
E9, c.1420C > T	p.Arg474*	Nonsense	F	13	NA	Strabismus	+	+/+	N	13 months	42 months	First words at 4.5 years	ADHD, aggressive, teeth grinding; mouths objects
E9, c.1420C > T	p.Arg474*	Nonsense	F	5.25	+	Strabismus	+	+/+	18	23	not yet	No words	Stereotypic outbursts
E9, c.1543C > T	p.Arg515*	Nonsense	F	51	+	Optic atrophy	+	+/+	NA	NA	No	Not able to speak, but uses sign language	Normal behavior, ID, cognitive abilities gradually deteriorated
E10, c.1603C > T	p.Arg535Ter	Nonsense	M	3.25	+	Strabismus	+	+/+	8	NA	Unable to walk	Lots of noises but no words	NA
E10, c.1603C > T	p.Arg535Ter	Nonsense	M	14	+	NA	-	-/+	15	NA	not walking	Moderate; Single words at 14 years	Aggressive outbursts, self-harm
E11, c.1801C > T	p.Gln601Ter	Nonsense	M	6.2	+	FEVR	+	+/NA	NA	NA	not yet	Says Mom and Dad with understanding, uses Makaton, points to body parts	Occasional temper; can bite others and self; repetitive movements
E12, c.1923dupA	p.Glu642Argfs*6	Frameshift	M	8.5	+	Strabismus, hyperopia	+	+/+	NA	14	8 years	Severe; few single words, gestures	Good social interaction, outburst of temper tantrums or crying, self/biting
E12, c.1925_1926del	p.Glu642Valfs*5	Frameshift	F	14.2	+	Strabismus	-	-/+	not yet	NA	No	Moderate, first words at 6 years; not speaking in sentences	Rages and tantrums, friendly personality, short attention span and poor eye contact; autism
MODERATE PHENOTYPE													
E3, c.99_100delTG	p.Gly34Asnfs*15	Frameshift	M	5.5	+	Strabismus	+	+/+	14; still head-leg	25	6 yrs (cannot stand alone)	Short sentences	Social and friendly boy; no behavioral problems; concentration is limited; sensitive to noises
E3, c.232C > T	p.Gln78*	Nonsense	M	11	+	Strabismus, hyperopia, astigmatism	-	+/+	NA	Didn’t crawl	3 yrs; at 11 yrs coordination problem	Unclear speech; at 11 yrs regression	Temper tantrums, aggression, frustration, anxiety, friendly personality, stereotypic movements
E4, c.283C > T	p.Arg95*	Nonsense	F	4	+	Normal	+	+/+	NA	12	4 yrs	Speech apraxia, ~50 words	When young, biting, banging the head in the wall, this has improved now
E5, c.705dupA	p.Gly236Argfs*35	Frameshift	F	14	+	Strabismus	+	+/-	12	NA	4.5 yrs	Babbling at 3 yrs, 14 yrs speaking simple sentences, read simple words	Autism, IQ = 65
E7, c.925C > T	p.Gln309*	Nonsense	M	4.5	+	Hyperopia	+	+/+	18	NA	walking at 4.6 (short distances)	Started speaking at 4 years, articulation was poor and hard to understand	Happy personality
E7, c.998dupA	p.Tyr333Ter	Nonsense	F	9	NA	NA	-	+/+	14	NA	4.2 (still had difficulties)	First words age 4; more fluent speech age 6; said to be 3 years behind with verbal skills	Violent outbursts associated with difficulty expressing emotions
E7, c.999C > G	p.Tyr333Ter	Nonsense	F	27	+	NA	-	+/+	30	NA	4.5 (ataxic)	First words at 4.5 years; can speak in sentences but speech very unclear	Aggressive, temper tantrums, self-injurious (biting, picking)
E7, c.1038_1044delGCTATCTinsGCT	p.Val349AlafsTer9	Frameshift	F	11	+	Strabismus, hypermetropia	+	+/+	NA	13.5	3.5 (ataxic)	Single words at age 5 years, talks in sentences at age 11 years	Stereotypies (clapping repeatedly, temper tantrums, aggressive to family)
I7, c.1081 + 1G > C	IVS6 Intron 7	Splice	M	3.8	+	Normal	+	+/+	not yet	not yet	not yet	Unclear speech	Good social interaction, very happy personality
E10, c.1612C > T	p.Gln538Ter	Nonsense	F	4.5	+	Strabismus	+	+/+	23	NA	2.5–3 years	First words at 3.4 years	Autism
333 kb incl. entire gene and ex. 35–37 of ULK4	Gross del	None	F	5.2	+	Hyperopia	+	+/-	14	NA	4.5 years (ataxic)	At 4.5 years could combine several words, count to 10	Friendly, social, short focus
505 kb incl. entire gene	Gross del	None	M	3	+	Esotropia	+	+/-	NA	not yet	not yet	Babbles and say “mama” and “dada”, before age 3 years	Happy, good eye contact
MILD PHENOTYPE													
E13, c.1981C > T	p.Arg661Ter	Nonsense	F	9.2	+	NA	+	+/-	11	NA	2.5	First words at 3,4 years	Obsessional behavior; dyspraxia
E13, c.2038_2041dup	p.Ser681*	Nonsense	F	13.2	+	Strabismus, myopia	-	+/+	12	NA	1.5	Mild, full sentences, but delayed	Social, autism, aggressive behavior, ADHD
E13, c.2038_2041dup	p.Ser681*	Nonsense	F	11	+	Strabismus, myopia	+	+/+	12	17	2	Mild, full sentences, but delayed	Communicative, social, aggression, ADHD
NORMAL PHENOTYPE													
E14, c.2128C > T	p.Arg710Cys	Missense	M	NA	normal	FEVR	NA	Normal	Normal	normal	normal	normal	Normal
E15, c.2142_2157dup16	p.His720*	Nonsense	M	NA	normal	FEVR	NA	Normal	Normal	normal	normal	normal	Normal

**Table 6 ijms-23-12564-t006:** Phenotype–genotype analysis (*n* = 35).

**Locus**	**Mutation Type**	**Number**	**Clinical Outcome**	**Remarkable Phenotypes**	**References**
Intron 5	Splice	3	Severe	Facial dysmorphisms, small head/microcephaly, axial hypotonia, peripheral spasticity, optical alterations, absent speech, severe ID, no eye contact, behavioral difficulties	Verhoeven et al. 2020 [1], Wang et al. 2019 [29]
Exon 6	Nonsense, frameshift	2	Severe	Facial dysmorphisms, small head/microcephaly, axial hypotonia, peripheral spasticity, severe speech impairment (no speech/2 words), severe ID, behavioral difficulties	Kuechler et al. 2015 [2], Kharbanda et al. 2017 [30]
Exon 4	Frameshift	1	Moderate-Severe	Facial dysmorphisms, microcephaly, axial hypotonia, peripheral spasticity, no walking, no speech (few words), autistic behavior	Kuechler et al. 2015 [2]
Exon 8	Missense	1	Moderate-Severe	Facial dysmorphisms, epilepsy microcephaly, axial hypotonia, peripheral spasticity, delayed walking, impaired speech (words not intelligible), ID	Kuechler et al. 2015 [2]
Exon 9	Nonsense	3	Moderate-Severe	Facial dysmorphisms, small head/microcephaly, optical alterations, axial hypotonia, peripheral spasticity, severe speech impairments (no speech/single words), severe ID and can have behavioral alterations	Kuechler et al. 2015 [2], Ligt et al. 2012 [4], Tucci et al. 2014 [28], Kharbanda et al. 2017 [30]
Exon 9	Frameshift	3	Moderate-Severe	Facial dysmorphisms, small head/microcephaly, optical alterations, axial hypotonia, peripheral spasticity, moderate speech (can have understandable words, can repeat short sentences), absent walking and speech, and can have behavioral difficulties	Kuechler et al. 2015 [2], Ligt et al. 2012 [4], Tucci et al. 2014 [28], Jin et al. 2020 [5], Kharbanda et al. 2017 [30]
Exon 10	Nonsense	2	Moderate-Severe	Facial dysmorphisms, small head/microcephaly, impaired speech (noises, few words), moderate/severe ID, behavioral difficulties	Kharbanda et al. 2017 [30]
Exon 11	Nonsense	1	Moderate-Severe	Facial dysmorphisms, optical alterations, axial hypotonia, unable to walk, impaired speech (few words), severe ID, behavioral difficulties	Kharbanda et al. 2017 [30]
Exon 12	Frameshift	2	Moderate-Severe	Facial dysmorphisms, optical alterations, peripheral spasticity, impaired speech (few words), ID	Kuechler et al. 2015 [2]
Exon 3	Nonsense, frameshift	2	Moderate	Facial dysmorphisms, optical alterations, axial hypotonia, peripheral spasticity, impaired speech (short sentences, unclear speech), ID	Winczewska-Wiktor et al. 2016 [23], Kuechler et al. 2015 [2]
Exon 4	Nonsense	1	Moderate	Facial dysmorphisms, microcephaly, axial hypotonia, peripheral spasticity, delayed walking, impaired speech (at 4 and 5 years < 50 words), ID, autistic behavior	Kuechler et al. 2015 [2]
Exon 5	Frameshift	1	Moderate	Facial dysmorphisms, strabismus, microcephaly, axial hypotonia, peripheral spasticity, delayed walking, impaired speech (simple sentences, read simple words), mild ID, and autistic behavior	Tucci et al. 2014 [28]
Exon 7	Nonsense, frameshift	4	Moderate	Facial dysmorphisms, axial hypotonia, peripheral spasticity, delayed walking (average 4,2 years), impaired speech (started speaking in 4 years, speaks in sentences, articulation poor), and can have behavioral difficulties	Ligt et al. 2012 [4], Tucci et al. 2014 [28], Kharbanda et al. 2017 [30]
Intron 7	Splice	1	Moderate	Facial dysmorphisms, microcephaly, axial hypotonia, peripheral spasticity, absent walking, impaired speech, ID	Kuechler et al. 2015 [2]
Exon 10	Nonsense	1	Moderate	Facial dysmorphisms, microcephaly, axial hypotonia, peripheral spasticity, impaired speech, autism	Kharbanda et al. 2017 [30]
Entire gene	Gross deletion	2	Moderate	Facial dysmorphisms, optical alterations, microcephaly, axial hypotonia, impaired walking, impaired speech, ID	Dubruc et al. 2014 [41], Kuechler et al. 2015 [2]
Exon 13	Nonsense	3	Mild	Facial dysmorphisms, axial hypotonia, delayed walking (before 24 months), mild ID, behavioral alteration (autism and aggression)	Kuechler et al. 2015 [2], Kharbanda et al. 2017 [30]
Exon 14 and 15	Missense and Nonsense	2	Normal	Normal phenotype with only optical alterations (FEVR)	Panagiotou et al. 2017 [35]

## Data Availability

Not applicable.

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
