# Peer review of "Correlation between Phenotype and Genotype in CTNNB1 Syndrome: A Systematic Review of the Literature"

_ijms, 2022, doi:10.3390/ijms232012564_

Round 1

Reviewer 1 Report

In this review, Miroševič et al. have systematically summarized current publications regarding to the genetic and clinical presentations of patients with CTNNB1 syndrome. They correlated the genotypes with phenotypes of patients and generated some informal insights.  These results are interesting and the literature search strategy and inclusion criteria are logical. Some minor points need to be addressed.

1.      Please clarify what body portion in patients MRI scan in the page 4 line 150. Is that brain scan? Line 151 mentioned that MRI is “normal” in 83.3% of cases, which seems conflict with the result in table 1. Should it be “abnormal”?

2.      It would be informative to show the percentage of each phenotype category in the current analysis of cases.

Author Response

We thank the reviewer for reading our manuscript and to the helpful comments and suggestions in improving our manuscript. We have revisited the manuscript and believe we were able to incorporate the suggestions successfully. Please find our point-to-point response showing how we have dealt with each of the comments that the reviewer noted.

  1. Please clarify what body portion in patients MRI scan in the page 4 line 150. Is that brain scan? Line 151 mentioned that MRI is “normal” in 83.3% of cases, which seems conflict with the result in table 1. Should it be “abnormal”?

Indeed, Magnetic resonance imaging is a brain scan. For clarification, please look at page 4, lines 492-494: »Magnetic resonance imaging (MRI) results were available for 24 cases of which 20 reports were normal (83.3%)”. In Table 1, we wrote that there were 24 valid cases, which means that the data were available only for this part of the included cases. Of those, 4 cases had an abnormal result (16.7%). We hope that this clarifies the misunderstanding.

  1. It would be informative to show the percentage of each phenotype category in the current analysis of cases.

Our Table 1 provides a summary of clinical features categorized by prevalence. This table has been improved (percentage of every phenotype for each category is shown in the parentheses). We have also added two figures to supplement this presentation. Figure 2 shows the clinical manifestations categorized by the primary and secondary criteria. Figure 3 shows the percentage of facial dysmorphism, ocular abnormalities, language development, and behavioral problems noted in reported CTNNB1 patients. We thank the reviewer for his helpful comments.

Reviewer 2 Report

Mutations in the CTNNB1 gene have been associated with neurodevelop-105 mental disorders, 

with cases of intellectual disability and speech delay. This is not a too rare form of clinical presentation but genotype/phenotype correlations are scarce.

This work present provides a systematic analyses of previously reported cases in which to analyze the prevalence of clinical manifestations, and to classify mutations according to their type, location, and disease severity.

This is a timely paper and I have few minor concerns:

- the authors should improve the general readability of the paper by using figures and pie charts detailing clinical manifestations

- attention to style and grammar is requested

- the authors should explain why 1025 records were excluded in their PRISMA flow diagram

- the criteria to categorize patients into 5 groups with respect to phenotype severity are not discussed

- the missense mutations were all class 4 and 5 in the ACMG scores or also class 3? This should be illustrated

Author Response

We thank the reviewer for his effort in improving our manuscript. We have revisited the manuscript and believe we were able to incorporate the suggestions successfully. Please find our point-to-point response showing how we have dealt with each of the comments that the reviewer noted.

- the authors should improve the general readability of the paper by using figures and pie charts detailing clinical manifestations

We thank for this important suggestion. We added two figures. Figure 2 shows the clinical manifestations categorized by the primary and secondary criteria. Figure 3 shows the percentage of facial dysmorphism, ocular abnormalities, language development, and behavioral problems noted in reported CTNNB1 patients. We thank the reviewer for his helpful comments.

- attention to style and grammar is requested

The paper was thoroughly reviewed again.

- the authors should explain why 1025 records were excluded in their PRISMA flow diagram

We added this explanation to Figure 1: Flow diagram of a systematic review. The majority of the excluded articles at this stage were review articles/meta-analysis, articles on cell and animal models, or article on CTNNB1 gain-of-function related to tumorigenesis.

- the criteria to categorize patients into 5 groups with respect to phenotype severity are not discussed

This is correct. We divided patients into 5 groups according to the severity of the phenotype. Because this is the first attempt to perform a genotype-phenotype analysis, there is no previous literature on this rare syndrome, so we could not refer to the existing literature. Therefore, categorization into groups was based on available questionnaires assessing developmental milestones (e.g., speaking, walking). Categorization was mainly based on patients' eye contact (present/absent), speech (Viking Speech Scale, Pennington, 2010), and cognition (ID; present/absent). Motor development was assessed only for normal and mild phenotypes, because motor development was less related to phenotype/symptom severity in the moderate and severe groups. The classification into five severity groups based on phenotypic and symptomatic characteristics formed the basis for the assessment of the correlation between genotype and phenotype. For an explanation, see Methods. See p. 22, lines 1572-1582.

- the missense mutations were all class 4 and 5 in the ACMG scores or also class 3? This should be illustrated.

This information was available only for 3 out of 6 missense cases. We added this information to the text (p. 8, lines 786-791): For all six missense mutations, American College of Medical Genetics and Genomics (ACMG) scores were extracted from both ClinVar and the Human Gene Mutation Database (HGMD). This information was available for 3 of 6 cases. Mutations c.1163T>C and c.2128C>T are classified as mutations of 'Uncertain Significance' and mutation c.1723G>A is classified as 'Pathogenic/Likely Pathogenic'.

We thank for this very timely comment and hope that our work will contribute to more available data in the further evaluation of the ACMG criteria for the CTNNB1-related mutations.